# The Hidden Potential of Informal Urban Greenspace: An Example of Two Former Landfills in Post-Socialist Cities (Central Poland)

Andrzej Długoński [1,*] and Diana Dushkova [2]

1 Institute of Biological Sciences, Cardinal Stefan Wyszynski University, Wóycickiego 1/3, 01-938 Warsaw, Poland
2 Landscape Ecology and Biogeography Lab, Department of Geography, Humboldt University Berlin, Rudower Chaussee 16, 2489 Berlin, Germany; diana.dushkova@geo.hu-berlin.de
* Correspondence: a.dlugonski@uksw.edu.pl; Tel.: +48-663-904-540

**Abstract:** The present study described analyses of two similar informal recreational green areas (former constructional waste disposal landfills) in two large cities (Warsaw city and Łódź city). On the basis of local society's opinions, the land use conditions related to current accessibility, management, safety, cleanliness, variety and diversity of facility and vegetation, neighbourhood and connections with the urban green infrastructure of the given sites were studied. Overall feedback posted by the site users indicated that, despite temporary land use, both sites are good leisure areas that provide cultural ecosystem services to the citizens. However, their undefined development makes them to varying degrees neglected and risky spaces, especially for the local community. The reflection of the results of the present study may help the local authorities to manage the spaces of former landfills in accordance with the needs of the local society as well as define new functions of informal urban green space in the sustainable spatial policy in post-socialist cities in Poland and Eastern Europe.

**Keywords:** abandoned degraded areas; old waste construction sites; recreational development; online survey; Warsaw city; Łódź city





## 1. Introduction

In the literature on the subject of landscapes, degraded areas are termed as "abandoned landscapes", "derelict landscapes", "lost spaces", "non-places", "anti-spaces", "brownfields" (industrial wasteland), "informal urban green space", "previously used land", etc. The common feature of such landscapes is that they are formed in the course of human activities because of rapid urbanisation, which leads to environmental degradation [1–8]. Pits, dumps and former landfills have become an integral part of the landscape and often accompany leisure areas, even in those used for short-term leisure [6,9–14]. Initially, finding harmony and order in such landscapes is difficult; however, with time, these landscapes are subjected to numerous transformations that make them suitable as recreation areas, particularly for local residents [15]. The fact that we choose our nearest vicinity as the location for short-term leisure has already been described in the literature, and such locations are dubbed as our "second homes" [15,16]. This phenomenon can be observed when examining the activities undertaken by residents of Poland's large cities, for whom, in terms of recreation, suburban areas are a generally accessible and inclusive alternative to sports clubs, especially in lockdown regimes [17,18]. Hallmann et al. [19] clearly indicated that such places are necessary not only to motivate local citizens to engage in physical activity but also to strengthen social bonds between them. It is precisely the above criteria of accessibility and inclusiveness that are met by two sites: Górka Kazurka in Warsaw and Górka Rogowska in Łódź. These areas are most frequently visited by local residents rather

than inhabitants of more distant districts and cities. Both places are attractive due to the benefits of cultural ecosystem services they offer in terms of physical activity [20–23]. It should be emphasised that both sites, despite being hills of modest height, tower above the vicinity, thus providing excellent vantage points [6,24].

Both hills were formed on former municipal landfill sites and currently belong to accompanying green areas—a type of open area covered by spontaneous vegetation [6,8,25–27]. They are developed primarily through the growth of plants and provide space for active leisure and sports events such as cycling races [28]. We may thus consider such areas as multi-functional spaces as an important element of urban green infrastructure [22,26,29]. A significant number of degraded sites are also present in those open suburban areas that will not be used as a location of housing or commercial development in the nearest future [14,30–32]. For some time, such sites will function as open spaces available for development by the local government. For example, the authorities of Washington, DC, earmarked degraded areas as the site of housing or commercial development [33]. Land development plans for Warsaw and Łódź specify that degraded areas will be allocated to parks and recreational facilities [22,34,35]. The approach used by the authorities of both cities seems to be rational as it will reduce construction pressure and address the leisure needs of residents [36,37].

As indicated in many studies, appropriate regeneration, renewal and adaptation of facilities to the needs of their users ensure the success of green areas in cities [14,38–41]. We may therefore posit that suitable development will also affect the attractiveness of degraded areas designated as leisure sites. An important aspect which should be emphasised in the present study is that leisure sites located within the administrative borders of a city are an immensely important enclave in the lives of their users, which is especially true for those who reside in larger cities in neighbourhoods such as those formed by multi-occupied buildings. In recent years, the awareness of the Europeans, including Poles, has significantly changed in the hierarchy of values. Apart from those strictly materialistic ones, arising from the necessity to secure basic existential needs, there appeared requirements related to self-realisation, access to culture, leisure and entertainment. A significant shift in the role played by sports in life has also been observed [42]. Constantly increasing living standards and relevant social campaigns have contributed to greater attention being paid to maintaining a healthy lifestyle. An appropriate duration of physical activity is an indispensable part of a consciously selected healthy lifestyle.

Regular exercise performed appropriate to age is a factor that reduces the risk of cardiovascular conditions, lowers cholesterol level and is beneficial to mental health [43–45]. Vert et al. showed the significance of physical activities undertaken in green areas (e.g., parks) to human health [41]. The researchers demonstrated a 7.3% reduction in death and a 6.2% decrease in the incidence of diseases among people who took advantage of a riverside park in Barcelona. This obviously translated into a significant reduction in funds allocated to healthcare and a positive impact on the economy. The authors of the work also stressed that a suitably prepared regeneration of green areas and their adaptation to various forms of physical activity is the key to use such locations not only for transit purposes but also for leisure. We may therefore assume that regeneration and adaptation of degraded areas to the requirements of recreation is a similar process.

The present study aimed to assess the recreational role of degraded areas at former municipal waste landfills (case studies) which function as undeveloped open public spaces. The assessment was made on the basis of the feedback of users; this approach is rarely used in Polish studies and is an important instance of urban ecology and landscape architecture research work conducted using this type of online survey.

This paper asks the following six main research questions:

1.　Why do city dwellers choose informal urban green spaces over landscaped parks in cities? Why exactly are these areas undeveloped?
2.　What kind of leisure activities dominate in selected degraded areas?
3.　Are the informal development facilities safe for users?

4. How are the analyzed facilities used by the local community?
5. What is the accessibility of the development sites and how are they connected to the city?
6. Are the facilities well managed and how are they maintained? Do visitors care about cleanliness?

## 2. Materials and Methods

The case study focused on degraded areas with similar history for selected former construction waste landfills in two large cities in Central Poland, specifically in the suburbs of Warsaw and Łódź (Figures 1 and 2). Górka Kazurka is situated in the northern part of the city in the Wyżyny neighbourhood in the district of Ursynów, which is the most populous district of Warsaw (Figure 3). It constitutes part of the Park Cichociemnych Spadochroniarzy AK and is also referred to as "Górka Trzech Szczytów". Górka Rogowska is situated in the northern part of Łódź in Bałuty district, which, like Ursynów, is the part of the city with the largest population (Figure 4). Both sites emerged in the 1970s as landfills for dumping construction materials from newly built neighbourhoods made using the large panel system (LPS) technique. The material was then covered with earth from excavations performed under the blocks or, as was the case in Warsaw, from metro line excavations [6,26,27,46]. The total area of the Górka Kazurka landfill is 10 hectares, of which 5 hectares form the forefield. Górka Rogowska is three times as large as Górka Kazurka; the forefield constitutes 20 hectares of its total area, and the hill proper has the size of 12 hectares. Both sites are adjacent to wooded areas. Górka Kazurka neighbours the fully protected Kabaty urban forest nature reserve, whereas Górka Rogowska adjoins a partly protected Łagiewniki urban forest. The feature which differentiates the two sites is the type of nearby settlements. Górka Kazurka is situated adjacent to large residential areas with multi-occupied buildings (the Wyżyny neighbourhood, which neighbours the site, alone has 10,000 inhabitants). In contrast, Górka Rogowska is located in the vicinity of single-family houses. It should be noted that Górka Kazurka is only sparsely overgrown with shrubs and small trees, while Górka Rogowska is almost entirely covered with trees and shrubs. In both cases, vegetation is ruderal, but at Górka Kazurka it is regularly mowed, and thus those plant communities are not visible there as in the Górka Rogowska site. Figures 1–3 show the location of the sites discussed in the present study.

The research methodology of the paper is presented in Figure 4.

The material used in this study included planning studies for the local level of the commune's spatial policy: for Warsaw, Land Use Planning Study for the City of Warsaw [34]; for Łódź, Land Use Planning Study for the City of Łódź [35], as well as thematic studies in urban ecology, environmental science and spatial planning [48].

Leisure function was measured using a modified quality assessment framework for green spaces and other open spaces. It is commonly used in the United Kingdom as part of the "Natural England's Country Parks Accreditation" [29,49]. An unquestionable advantage of the method is that it constitutes a standard tool, is readily applicable and offers clear assessment criteria (5 credits, where 5—excellent, 4—good, 3—fair, 2—poor and 1—very poor). It is based on visitor feedback, which is quite important in terms of the practical use of the area. It may also be used in other case studies and assessed as part of a public consultation. This universal analytical framework is related to (1) welcoming place (physical access and provision of signage and information); (2) healthy, safe and secure (facilities and opportunities offered for exercise, as well as general safety and security); (3) clean and well maintained (litter and waste management and grounds maintenance); (4) conservation and heritage (natural or historic heritage, environmental designations, information about ecology and range of natural features); and (5) diversity and variety (range of facilities and opportunities for activities) of the given sites. A modification introduced for the purpose of this study involved the omission of the "heritage" category due to negligible historical and conservation value of both analysed sites. The "linking

and neighbourhood" category was also added, as both hills are former landfill sites, and because they are situated in lowlands, they constitute landmarks in urban space.

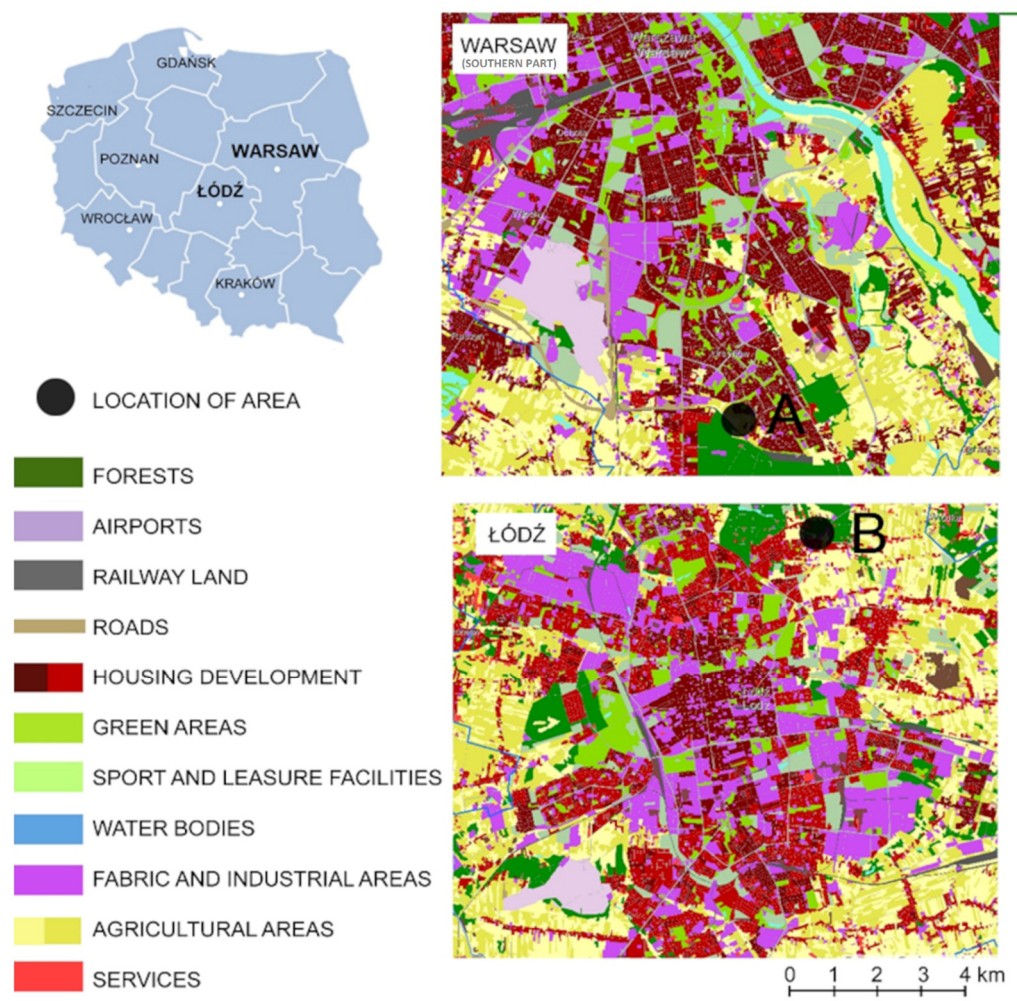

**Figure 1.** Location of the investigated degraded sites: (**A**) Górka Kazurka; (**B**) Górka Rogowska. Authors' own elaboration based on [47].

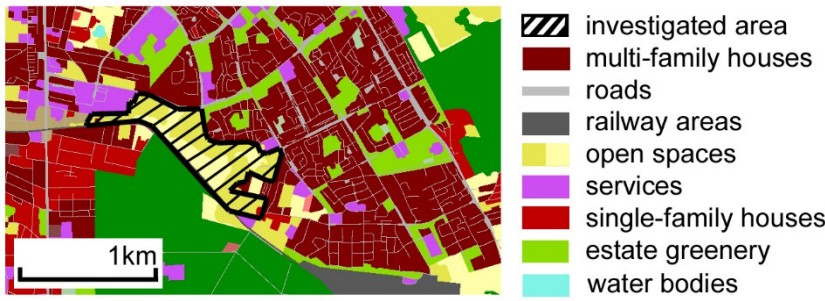

**Figure 2.** Location of Górka Kazurka in the suburbs of Warsaw, Poland. Authors' own elaboration based on [47].

The analysis of user ratings was based on the feedback of local society members. Users of the online survey can rate a given location by awarding a score of 1 to 5 credits. They can also post their feedback on the value of the place, its advantages and disadvantages by writing their own comments. This study analysed feedback both with and without users' comments: the latter contains only numerical ratings. Feedback without comments was treated as the site's overall rating. Subsequently, users' comments were assigned to

categories being analysed. Table 1 lists the total number of ratings, grouped into those with and without comments.

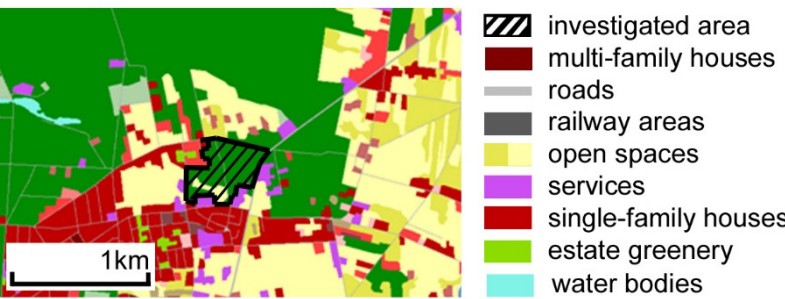

**Figure 3.** Location of Górka Rogowska in the suburbs of Łódź, Poland. Authors' own elaboration based on [47].

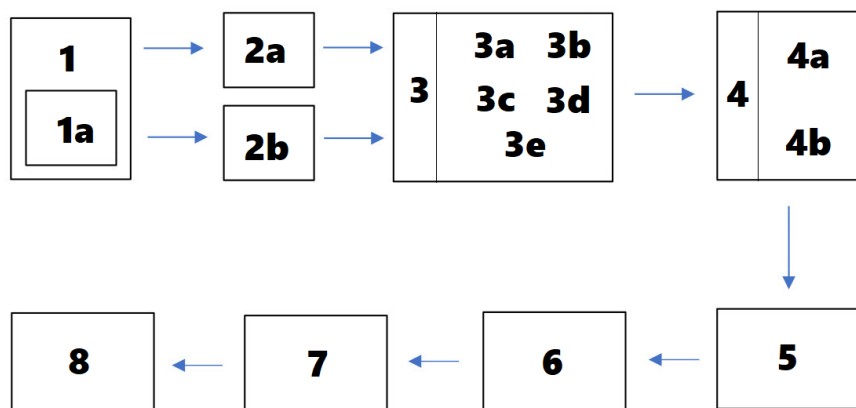

**Figure 4.** Research methodology. **1a**—definition of the research problem; **1b**—selection and justification of case studies; **2a** -definition of the aim, scope and subject of the article; **2b**—determination of research questions; **3**—cameral and analytical-statistical studies; **3a**—literature query for methods of assessing open public spaces, green areas and degraded areas [3–9,13,15,16,24–29,32,33,38–40,48–77]; **3b**—literature query for informal urban green spaces [2,10–12,14]; **3c**—collection of input materials concerning the study area [1,6,26,27,34,35,46]; **3d**—determination of research method/research analytical framework; **3e**—development of an evaluation questionnaire form for case study (Appendix A: Figure A1); **4**—field studies; **4a**—photographic documentation of the study site; **4b**—distribution of an evaluation questionnaire form sheets among the site visitors (online forms, written notice posted on the surveyed sites); **5**—collection and compilation of the study site evaluation statistics; **6**—elaboration of results; **7**—discussion of results; **8**—conclusions, guidelines and summary.

**Table 1.** Number of users who rated the areas.

|  | Górka Kazurka | Górka Rogowska |
|---|---|---|
| rating with comments | 150 | 78 |
| rating without comments | 230 | 96 |
| total number of ratings | 380 | 174 |

Next, according to the method presented in Natural England's Country Parks Accreditation, a weighted mean was calculated for the total rating in a specific category. The highest weight was 5, and the lowest was 1. The following method was applied to individual ratings:

$$( n_{oc} \times n_{pt} )$$

where:

$n_{oc}$—number of individuals who rated the site by providing a specific number of credits; $n_{pt}$—number of credits in the entire category,

Resulting in the following formula:

$$\frac{(n_{oc} \times 5) + (n_{oc} \times 4) + (n_{oc} \times 3) + (n_{oc} \times 2) + (n_{oc} \times 1)}{N}$$

where *n* is the total number of ratings in a specific category.

Weighted means calculated for each category were multiplied by 2, as the method reported in Natural England's Country Parks Accreditation requires the conversion of the score so that the maximum equals 10. Results for all categories were subsequently totalled, and the overall assessment of the analysed sites was performed according to the results.

As in the case of individual components, the overall rating of the sites was based on a five-credit scale. A total of 60 possible credits were divided into 5 tiers, thus enabling overall assessment of the sites.

0–12 credits: very poor
13–24 credits: poor
25–36 credits: fair
37–48 credits: good
49–60 credits: very good

## 3. Results

Assessment results are presented in Table 2. The table contains information on the frequency of feedback posted by site users. Note the disproportion in the number of ratings and comments, which in itself is information indicating differences between the two sites in terms of attractiveness.

**Table 2.** Analysis of site ratings.

| | Accessible and welcoming place | | | | | |
|---|---|---|---|---|---|---|
| | very poor n (%) | poor n (%) | Fair n (%) | good n (%) | excellent n (%) | Total n (%) |
| Górka Kazurka | 0 (0.0) | 1 (5.6) | 2 (11.11) | 6 (33.3) | 9 (50.0) | 18 (100.0) |
| Górka Rogowska | 0 (0.0) | 0 (0.0) | 3 (75.0) | 0 (0.0) | 1 (25.0) | 4 (100.0) |
| | Healthy, safe and secure | | | | | |
| | very poor n (%) | poor n (%) | Fair n (%) | good n (%) | excellent n (%) | Total n (%) |
| Górka Kazurka | 1 (5.3) | 0 (0.0) | 1 (5.3) | 1 (5.3) | 16 (84.1) | 19 (100.0) |
| Górka Rogowska | 0 (0.0) | 1 (16.7) | 5 (83.3) | 0 (0.0) | 0 (0.0) | 6 (100.0) |
| | Cleanliness and maintenance | | | | | |
| | very poor n (%) | poor n (%) | Fair n (%) | good n (%) | excellent n (%) | Total n (%) |
| Górka Kazurka | 0 (0.0) | 0 (0.0) | 4 (23.5) | 2 (11.8) | 11 (64.7) | 17 (100.0) |
| Górka Rogowska | 0 (0.0) | 1 (10.0) | 6 (60.0) | 2 (20.0) | 1 (10.0) | 10 (100.0) |

**Table 2.** *Cont.*

| | very poor n (%) | poor n (%) | Fair n (%) | good n (%) | excellent n (%) | Total n (%) |
|---|---|---|---|---|---|---|
| **Diversity and variety** | | | | | | |
| Górka Kazurka | 0 (0.0) | 0 (0.0) | 4 (5.2) | 23 (29.9) | 50 (64.9) | 77 (100.0) |
| Górka Rogowska | 0 (0.0) | 0 (0.0) | 5 (50.0) | 3 (30.0) | 2 (20.0) | 10 (100) |
| **Connections and neighbourhood** | | | | | | |
| Górka Kazurka | 0 (0.0) | 1 (5.3) | 2 (10.5) | 9 (47.4) | 7 (36.8) | 19 (100.0) |
| Górka Rogowska | 0 (0.0) | 0 (0.0) | 0 (0.0) | 13 (27.1) | 35 (72.9) | 48 (100.0) |
| **Overall assessment—without comments** | | | | | | |
| Górka Kazurka | 1 (0.4) | 3 (1.4) | 12 (5.2) | 13 (5.6) | 201 (87.4) | 230 (100.0] |
| Górka Rogowska | 1 (1.0) | 1 (1.0) | 1 (1.0) | 25 (26.2) | 68 (70.8) | 96 (100.0) |
| **Overall assessment—total rating** | | | | | | |
| Górka Kazurka | 2 (0.5) | 5 (1.3) | 25 (6.6) | 54 (14.2) | 294 (77.4) | 380 (100.0) |
| Górka Rogowska | 1 (0.6) | 3 (1.7) | 20 (11.5) | 43 (24.7) | 107 (61.5) | 174 (100.0) |

The "accessible and welcoming place" criterion was analysed as the first one. The total number of comments on this criterion for Górka Kazurka was 18; 50% of users felt that the place offers very good access. Another 33% thought that access to the site is good. Nearly 17% of users noticed certain inconveniences related to access to the site in terms of limited mobility opportunities for the disabled. The same assessment criterion for Górka Rogowska in Łódź revealed that as many as 75% of users were concerned about those inconveniences (Figure 5).

In terms of safety, almost 90% of reviewers rated Górka Kazurka as a safe place. They also appreciated the site's positive influence on health, emphasising that the site is very good for engaging in physical activity. In particular, the users frequently referred to the Kazoora Bike park situated on Górka Kazurka—the largest cycling park in Mazowieckie Voivodeship. For Górka Rogowska, 83% of users considered it relatively safe, and one person rated it as a dangerous place. The comments left by the users mentioned that the construction waste, the main material forming the hill, sticks out in some locations on the site (Figure 6). It was also pointed out that the site is a venue of meetings, during which illegal bonfires are made, alcohol is consumed and drugs are taken. Walking opportunities were considered to be a positive aspect.

In terms of cleanliness and maintenance of facilities, the reviewers mentioned litter present on the sites as well as equipment and its quality. A total of 76% of the users of Warsaw's Górka Kazurka wrote that the place is very clean and very well or well maintained. Others rated these aspects as satisfactory. The users noticed that unhardened

cycling routes and footpaths affect the enjoyment of the site, as after heavy rain, the area of the hill and the forefield becomes muddy. For Górka Rogowska, 60% of users stated that the site displays an average level of cleanliness and maintenance. According to 30% of users, the site is very clean and well maintained. One of the reviewers found it neglected and dirty. Users who rated Górka Rogowska as a relatively well-maintained place pointed out that the site contains scattered rubbish and that footpaths are not secured and are damaged by waste (Figure 7). There were also comments on the lack of care for vegetation as well as the absence of benches.

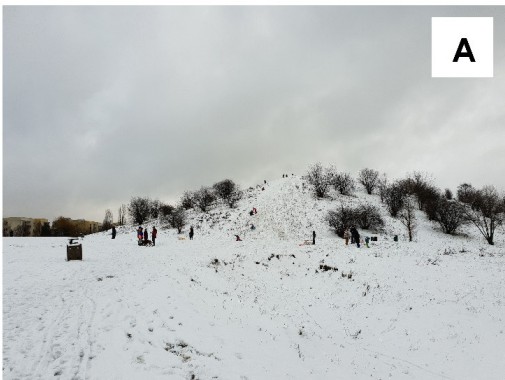
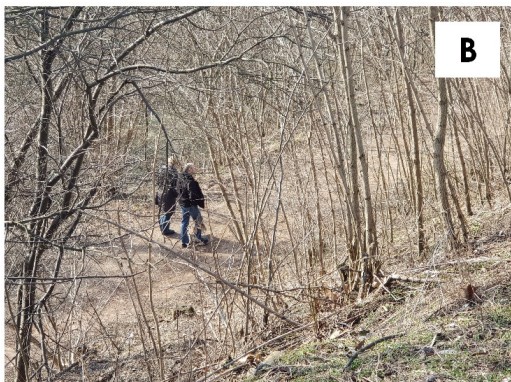

**Figure 5.** Good access to the site with limited mobility opportunities for the disabled in Górka Kazurka (**A**) and Górka Rogowska (**B**).

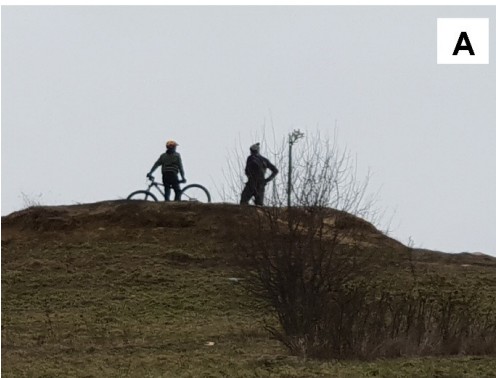
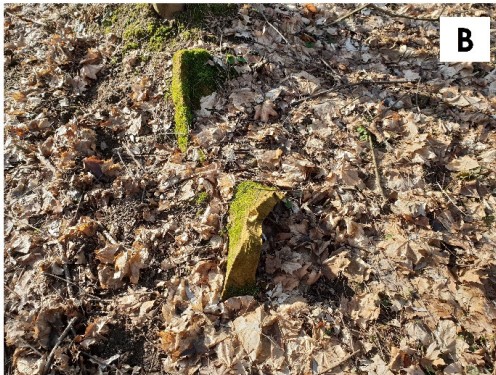

**Figure 6.** Safety of Górka Kazurka (**A**) bike park for creating physical activity vs. dangerous sticking out of construction waste in Górka Rogowska (**B**).

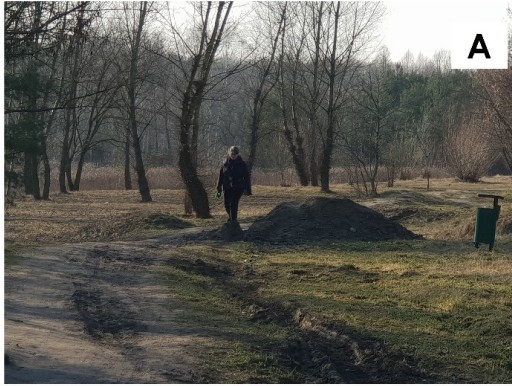
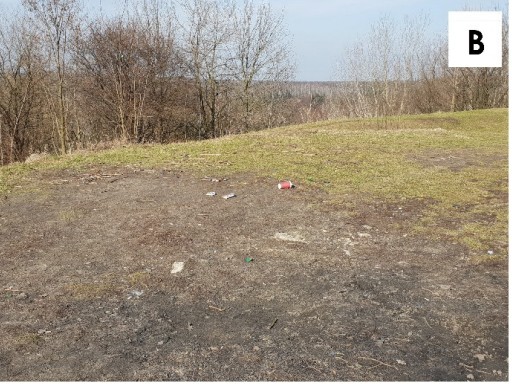

**Figure 7.** Cleanliness and maintenance—muddy footpaths in Górka Kazurka (**A**) and scattered rubbish in Górka Rogowska (**B**) influence users to think that the hills are neglected and dirty.

Considering the variety of available active and passive recreation facilities, nearly 95% of Górka Kazurka users concluded that the site is a very good (64.9%) or good (29.9%) place for leisure. They placed particular emphasis on cycling opportunities and facilities for children (the forefield features a skatepark). Comments also referred to the presence of an outdoor gym, a dog park and a picnic area. The site is also considered to be a very good place for walking, although due to the absence of vegetation offering cover, strolls may be less pleasant in windy weather. It was also indicated that the place is conducive to social integration because of occasional festivities organised at the foot of the hill or competitions held in the bike park. The array of attractions offered by Górka Rogowska is much more limited. Half of its users consider it a place adequate for active leisure, mainly walking (Figure 8). Others regard it as a good or very good place for various types of physical activity, which, besides walking, include jogging or cycling. Both the hill area and the forefield are overgrown with trees and shrubs, which according to users improves well-being and has a positive effect on the psyche.

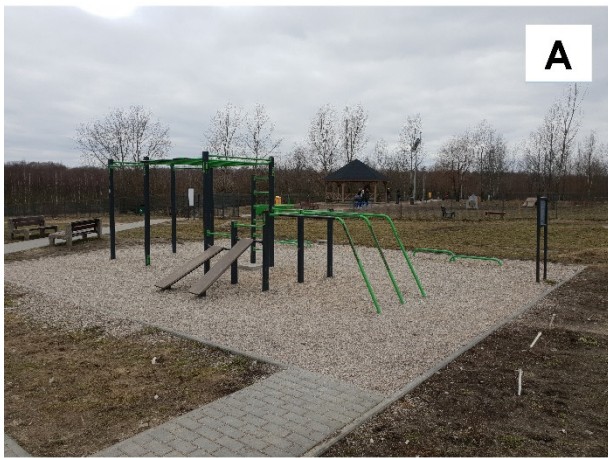
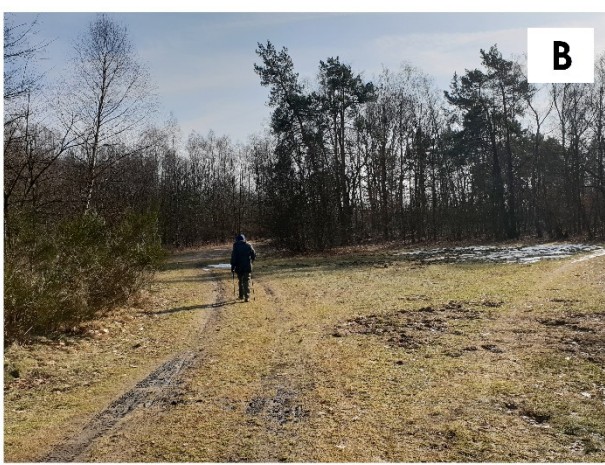

**Figure 8.** Górka Kazurka's (**A**) equipment—outdoor gym and dog park vs. Górka Rogowska's open spaces for walking (**B**).

Because both sites are located in the lowlands, they are good vantage points, a feature which is equally appreciated by users of Górka Kazurka and Górka Rogowska. However, it should be stressed that for Górka Rogowska, because of a limited number of attractions, the opportunity to admire the panorama of the surroundings was the main reason for visiting, as emphasised by reviewers in their comments (Figure 9).

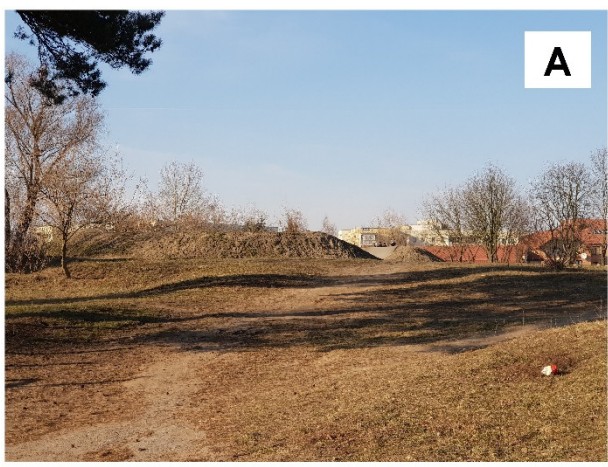
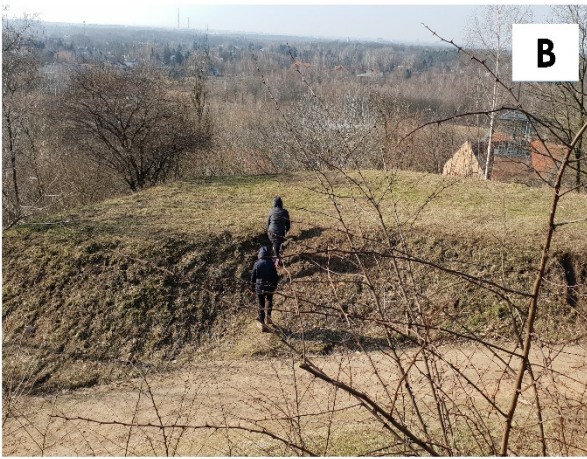

**Figure 9.** Site opportunity to admire the panorama of the surroundings in Górka Kazurka (**A**) and Górka Rogowska (**B**).

In both cases, the reviewers mentioned easy access to the sites. Access to Górka Kazurka was rated as very good: it is accessible by car, bicycle and public transport, and there are car parks available, with a direct connection to local pedestrian routes in the vicinity. The situation of Górka Rogowska is quite different—it is possible to access the site using public transport or by car; however, the hill is almost entirely obscured by the local school and vegetation (Figure 10).

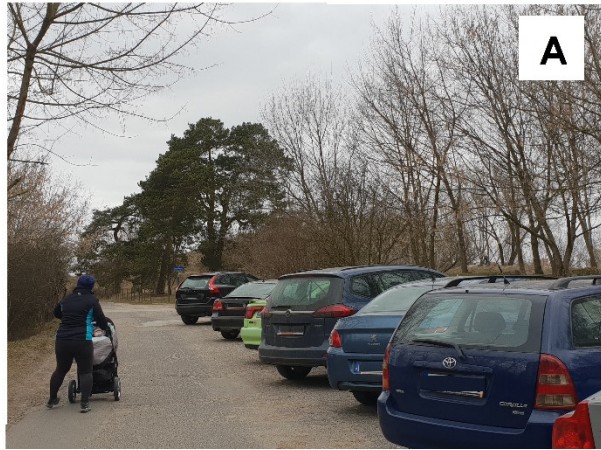 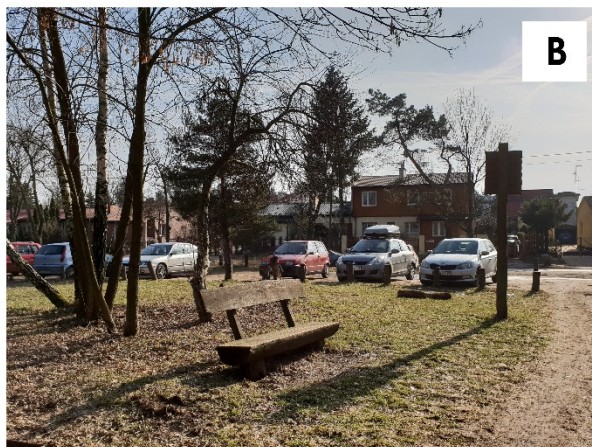

**Figure 10.** Access by cars is the best possibility to quickly and easily reach the sites from outside in Górka Kazurka (**A**) and Górka Rogowska (**B**).

Overall feedback posted for the sites by their users, both with and without comments, indicates that both sites are good or very good leisure areas. However, it should be noted that for Górka Rogowska, more attention should be paid to cleanliness and maintenance issues. It was felt that the site, despite its potential, is under-invested and neglected (Table 2).

The overall rating of the sites, calculated according to the method suggested in Natural England's Country Parks Accreditation, is presented in Table 3. It appears that users awarded a much better rating to Warsaw's Górka Kazurka. On the five-level scale, the users rated Górka Kazurka as a very good place for leisure, while the users of Górka Rogowska rated the site as a good place for leisure.

**Table 3.** Number of users who rated the areas.

| Aspect | Criterion | Counting Method | GK | GR | Total Credits Available |
|---|---|---|---|---|---|
| accessible and welcoming places | access arrangements; access on site; disabled access | Converted into score/10 | 8.4 | 7.0 | 10 |
| health, safety and security | site area; lighting; boundary maintenance; visibility; general safety | Converted into score/10 | 9.2 | 5.6 | 10 |
| cleanliness and maintenance | dog fouling; litter; number of bins; overall cleanliness; landscape maintenance; seating quality | Converted into score/10 | 9.0 | 6.6 | 10 |
| diversity and variety | provision of seating; provision of shelter; parking; cycle park; physical attributes; range of activities; provision of play facilities; provision of playing fields | Converted into score/10 | 9.2 | 7.4 | 10 |
| Connections and neighbourhood | links to surroundings links to the wider countryside and urban green infrastructure | Converted into score/10 | 8.4 | 9.4 | 10 |
| Overall assessment (rating with and without comments) | | Converted into score/10 | 9.3 | 8.9 | 10 |
| TOTAL | | | 53.5 | 44.9 | 60 |

GK—Górka Kazurka, GR—Górka Rogowska.

## 4. Discussion

　　Green areas, regardless of their origin (as natural places or those formed on degraded areas), are important leisure enclaves for residents of large cities [10,41]. They contribute to a heightened amount of physical exercise, which in the time of an increasingly sedentary lifestyle has a substantial impact on health [43]. Of particular importance is the positive influence of green areas on mental health [50].

　　Among the recognised methods on social, recreational and natural functioning of informal green spaces, there is a lack of evaluation methods on former urban landfills [1,6,24,26,28]. Therefore, to evaluate degraded areas functioning as informal green spaces, there is a need to develop methods that can evaluate various open spaces and public areas. The first group of analyses addresses all urban degraded areas, both built-up post-industrialised and lost or uninvested open spaces [51]. Most of the evaluation studies also discuss specific issues such as the accessibility assessments of these places [52], economic and recreational values [53–62] and only natural values or quality of the natural environment of abandoned urban spaces [1,27,63]. The second group of analyses concerns public spaces on the city scale, which might be used for comparative analyses of other similar areas or in the context of ecosystem services [6,21,22,64–66]. The third group of methods does not comprehensively analyse the issue of the leisure functioning of green spaces but focuses only on particular issues selected by the authors, such as management aspects, availability, attractiveness, equipment, safety, neighbourhood and characteristics of user groups [67–72].

　　The first multi-faceted assessment method of open spaces perception was described by Scott [73] with later modifications [74,75], referring to various key factors such as biological origins, sense of place, cultural associations, age, naturalness, familiarity, perceiving of landscape value, feeling evoked, sound and smell or viewer's background. However, it does not provide guidance on how to conduct such an overall multi-faceted assessment for the selected site. On the other hand, other methods of perceiving public places exclude user groups on the basis of different types of disability [67,76].

　　In the present study, the universal method of assessing open spaces was used [29,49]. This method was chosen because it broadly defines the aspects of the land-use conditions related to current accessibility/disability, management, safety, cleanliness, variety and diversity of facility and vegetation, connections and neighbourhood of given sites by using a rating scale for each criterion. Thus, this method is applicable to various types of open spaces, and it was used to assess degraded areas that are former constructional waste landfills in cities and are now functioning as informal green spaces in cities. The method applied here has been successfully used in the United Kingdom on multiple occasions [29,49]. It is a repeatable method that yields satisfactory results, and in contrast to other cited methods of selected issues on urban green spaces, it can easily be compared because of the use of a credit scale. Currently, user feedback is commonly used to evaluate leisure function in Western European countries. Such studies are exemplified by work performed by Hallmann et al. [19], who investigated the relationship between infrastructure and physical activity undertaken by residents. Their conclusions prove that creating inclusive, generally accessible spaces which enable engaging in sports that do not require considerable financial outlays (e.g., jogging) is of highest importance. Both sites discussed in this article meet the aforementioned criteria.

　　Schreerder [77] clearly indicated that among all European nations, Poland has the lowest percentage of individuals engaging in physical activity at least once a month. Only 45% of Poles admit that they engage in physical activity at least once a month, of which 45% perform physical activity outdoors, most often in a park. In light of the above finding, it seems reasonable not only to provide Poles with access to such places but also to ensure a possibly wide array of sports opportunities for individuals of various ages and limitations, e.g., disabilities.

　　The observations made in the present study indicate that users of Górka Kazurka and Górka Rogowska are comfortable engaging in physical activity on a site that was formerly

a landfill. On the contrary, in both cases, they regarded the sites as favourable leisure areas offering attractive vistas of the surrounding landscape. They, however, indicated certain imperfections of the areas, and such imperfections were more for Górka Rogowska. User feedback is a very important indicator for further development of both sites, which allows us to outline the direction of changes required for the sites to become even more attractive. It is also more important considering the fact that the sites border on nature reserves. Górka Kazurka borders the fully protected Kabaty urban forest, while Górka Rogowska adjoins Łagiewniki urban forest, which forms part of the Łódź Hills Landscape Park that is subject to partial protection. The purpose of protecting both woods is to maintain their basic natural resources. Thus, the adaptation of Górka Kazurka and Górka Rogowska to recreational needs might be aimed to limit anthropogenic pressure on neighbouring nature reserves [6].

During the research, it was found that the feedback provided by the reviewers clearly focuses on the needs of the visitors and the assessment of the equipment installed on the sites. Users paid close attention to issues that often elude researchers who evaluate the leisure function using methods based on site observation. For example, users raised the subject of the presence of litter in the area. An important point to consider is the fact that very often users spend more time or pay multiple visits to the area, which makes their subjective evaluation more credible than that of a researcher who spends merely several hours on the site. Users indicate that, although in good weather conditions the absence of trees is undoubtedly an advantage, in windy weather, the area does not offer any protection from the wind.

An online survey was used to assess the land-use conditions related to current accessibility, management, safety, cleanliness and variety and diversity of facility and vegetation, neighbourhood and connections with the surroundings of both sites. Although the online tool cannot replace field surveys, interviews or estimating the maximum resting capacity of sites, it offers the possibility of gathering information on a particular area [23] during pandemic regulations [17,18]. This is further confirmed by social researchers and psychologists who clearly show that in times of isolation, the Internet is the biggest source of information on recreation [52]. In contrast to a paper-pen and filled-in questionnaire, an online questionnaire does not cover all aspects assessed in a survey, and the users can choose to respond to only those questions that they wish to express their opinion. Nevertheless, it must be emphasised that the tool allowed us to reach a large number of respondents who freely voiced their views [52]. By awarding a certain number of credits (even without posting any comments), the users provided an overall rating of the sites, and their feedback deserves to be considered by researchers in their own assessments. It may complement site observation very well, and users' opinions may indicate shortcomings and outline the nature of changes that should be considered in development plans and directions for renewal.

## 5. Conclusions

According to planning studies, the areas of former landfills constitute a basis to construct open recreational and sports parks. However, they function as unmanaged open spaces with "informal" recreation arranged freely by the local community. Data collected from respondents' site observation show that the sites' facilities consist of benches, bins and information folders, with insufficient recreational and sports facilities such as recreational fields, gyms, picnic paddies and raincoats or decorative vegetation providing comfort and protection against adverse climatic conditions. Thus, in both cases, recreational infrastructure does not ensure proper functioning of the area. However, Górka Kazurka in Warsaw serves as a sports and recreational park operating throughout the year due to its proximity to multi-occupied housing estates and the lack of similar public areas in its vicinity. Moreover, the neighbouring Kabaty urban forest is strictly protected, and certain types of activities other than walking and cycling are not allowed. The situation in Górka Rogowska in Łódź is different: a similar undeveloped open area is used occasionally, as

residents of neighbouring single-family houses use their own home gardens, and weekend leisure is dominated by the adjacent Łagiewniki urban forest which is partly subjected to nature protection regulations.

An assessment of the leisure function of Górka Kazurka and Górka Rogowska sites was performed on the basis of online feedback from their users. The study revealed that both sites are characterised by considerable leisure potential mainly due to their hilly terrain. It was found that even the minimal makeshift and often quite provisional land development is sufficient to transform former degraded areas into attractive open public spaces that provide recreational attractions throughout the year. Thus, the mandatory requirement is to ensure availability of a given area (good access to the site and its facilities, multi-functional recreation and safety of the area) as well as good location and land use of the surrounding area (multi-occupied housing estates and lack of similar competitive areas in the vicinity). It is worth noting here that multi-functional development and full range of leisure facilities of a degraded area do not ensure the success of this place as an attractive recreational site visited throughout the year by a maximum number of users. However, it should be expected that in both cases, the construction of proper recreational infrastructure and the modernisation of current land development will increase the safety of the sites, thereby making them more attractive to users and attract potential users from remote parts of the city or new visitors.

The user opinions presented in the article may help the management (local authorities) to organise the spaces of former landfills and to manage these spaces for sport and recreation (passive and active recreation) in accordance with the needs of the local society in order to maximise the extraction of protected neighbouring urban forests from excessive anthropopressure, thus making full use of the potential of degraded sites.

The results of this paper may also be helpful in developing a concept for revitalising degraded areas, defining their new functions in the sustainable spatial policy of communes and providing premises for other cities in Poland and Europe.

**Author Contributions:** Conceptualization, A.D.; methodology, A.D.; software, A.D.; validation, A.D.; formal analysis, A.D.; investigation, A.D.; resources, A.D.; data curation, A.D.; writing—original draft preparation, A.D.; writing—review and editing, A.D. and D.D.; visualization, A.D.; supervision, A.D.; project administration, A.D.; funding acquisition, A.D. All authors have read and agreed to the published version of the manuscript.

**Funding:** This research received no external funding.

**Institutional Review Board Statement:** Not applicable.

**Informed Consent Statement:** Informed consent was obtained from all subjects involved in the study.

**Data Availability Statement:** The database for the online survey is given in the article. The sources of elaboration on the database were based on the visitors' opinions and ratings related to given case studies from an online forum available in the links listed below. Górka Rogowska user opinions (https://goo.gl/maps/Rc8PTEZmtDJRM5sJ7, accessed on 12 January 2021). Górka widokowa ''Rogi'' user opinions (https://goo.gl/maps/QPgF52gRLB2rp8Xs7, accessed on 12 January 2021). Bikepark Kazoora user opinions (https://goo.gl/maps/NpLgvbH5NRNQd4jL9, accessed on 12 January 2021). Wzgórze Trzech Szczytów user opinions (https://goo.gl/maps/ghGUgLcykiNe48aQ7, accessed on 12 January 2021). Park imienia Cichociemnych Spadochroniarzy user opinions (https://goo.gl/maps/UangnWeN6zR6GiQY7, accessed on 12 January 2021). All visitors' opinions from the above-mentioned online forum sources are automatically processed to detect inappropriate content and spam by the webmaster. Some opinions have been omitted in order to comply with web publishing rules and the law regulations. Hence, the greatest value to the development of the authors' database on investigated areas was the honest and objective content posted.

**Acknowledgments:** We thank Justyna Marchewka from the Department of Human Biology, Institute of Biological Sciences, Faculty of Biology and Environmental Sciences, Cardinal Stefan Wyszynski University in Warsaw, for help in statistical analysis.

**Conflicts of Interest:** The authors declare no conflict of interest.

## Appendix A

**Open space assessment note for site (name): _________________**

**№______________　　　　　　　　Date__________________**

*Dear Sir/Madam,*

*We kindly request you to complete this evaluation form. It consists of five sections (aspects), divided into thematic subcategories (criterion). The information obtained will be used to prepare a scientific study summarizing the recreational preferences of city residents.*

*The questionnaire should not take you more than 15 minutes to complete, and the information you provide will be an important guide for us and may, in the future, prove helpful in developing informal recreational places in cities.*

*We would like to thank you for your kind treatment of our survey.*

*Researchers*

| Aspect | Criterion | Score * | Visitor's comments |
|---|---|---|---|
| *Accessible and welcoming places* | Physical access and provision of signage and information:<br>- access arrangements<br>- access on site<br>- disabled access | | |
| *Health, safety and security* | Facilities and opportunities offered for exercise, as well as general safety and security:<br>- site area<br>- lighting<br>- boundary maintenance<br>- visibility<br>- general safety | | |
| *Cleanliness and maintenance* | Litter and waste management, grounds maintenance and management and maintenance of buildings:<br>- dog fouling<br>- litter<br>- number of bins<br>- overall cleanliness<br>- landscape maintenance<br>- seating quality | | |
| *Diversity and variety* | Range of facilities and opportunities for activities:<br>- provision of seating<br>- provision of shelter<br>- parking<br>- cycle park<br>- physical attributes<br>- range of activities<br>- provision of play facilities<br>- provision of playing fields | | |
| *Connections and neighbourhood* | Linkages:<br>- links to surroundings<br>- links to the wider countryside and urban green infrastructure | | |
| Overall assessment : | | | *(optional other comments)* |
| *Score:<br>5 - excellent, 4 - good, 3 - fair, 2 – poor, 1 - very poor. | | | |

**Figure A1.** Example of evaluation form for given case studies. Authors' own elaboration based on [29,49].

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
