# Peer review of "The Hidden Potential of Informal Urban Greenspace: An Example of Two Former Landfills in Post-Socialist Cities (Central Poland)"

_sustainability, doi:10.3390/su13073691_

Round 1
Reviewer 1 Report
The article deals with a very interesting and under-researched topic of leisure and green informal spaces. The study brings interesting conclusions relating to bottom-up urbanism. In my opinion, this topic is worth developing in further research, because we do not learn what this high level of safety results from (high level of social control related to frequency of use, lack of previous crime episodes, etc.).
The issue of the research technique used ("online feedback of local society members") needs to be better addressed. While the explanation of the use of this technique in the context of the pandemic is convincing, the paper should also indicate: (1) how users may have been exposed to the questionnaire (whether it was a website, what survey format was used, how the information was disseminated) (2) how the researchers estimate the relationship between the sample and the informal green spaces users population.
Author Response
The authors of this paper would like to thank the reviewers very much for their insightful review and suggestions. All responses were incorporated directly into the paper or below.
Ad.2. In pandemic time, it is difficult to estimate maximum resting capacity of given sites, and thus relationship between the sample and the informal green spaces users population. However, we assume that the large number of on-line forms (380 for Górka Kazurka and 174 for Górka Rogowska) of responses is meaningful to estimate the assessed aspects included in the evaluation of the study sites. However, we think that this issue is very important and should be the subject of a separate study.

Reviewer 2 Report
Dear authors,
The paper is interesting and original, as it provides insights into the Hidden Potential of Informal Urban Greenspace.
Below I would give some comments to better clarify the research path and structure:
- Introduction
The theoretical background is suitable for the aim of the paper, but I would suggest implementing the literature review on practices about Informal Urban Greenspaces. In briefly mentioning the main aim of the work, it could be useful to highlight the research question, research objectives, and the paper Sections for emerging the research structure. Please, I suggest eliminating brief paragraphs as 1.1 for better readability of the whole text.
- Materials and methods
Materials and methods could be better explained, I would suggest inserting a research methodology explanation specifying each research step able in responding to the research objectives (a graphic on research methodology approach could help). I would also suggest better highlight which research analytical framework (for example the “universal method of assessing open spaces”?) is used for the case study analysis. For every research step, please show tools and approaches used with literature references.
- Discussion
The findings and their implications should be discussed also with limitations of the work highlighted.
Author Response
The authors of this paper would like to thank the reviewers very much for their insightful review and suggestions. All responses were incorporated directly into the paper or below.
Ad.1. Implementing the literature review on practices about Informal Urban Greenspaces requires a separate literature review. Given the slimness of the editor's time allotted in the minor revision, we think that this issue is very impoertant and should be the subject of a separate study, as the second reviewer (a series of research articles on informal/ degraded green spaces) also partly suggests.
The evaluation of the study area was carried out according to the method presented in Natural England’s Country Parks Accreditation. This method as research framewrok was also more highlighted in article.
